# Genome-Wide Identification and Expression Analysis of *DWARF53* Gene in Response to GA and SL Related to Plant Height in Banana

**DOI:** 10.3390/plants13030458

**Published:** 2024-02-05

**Authors:** Ning Tong, Chunyu Zhang, Xiaoqiong Xu, Zhilin Zhang, Jiahui Li, Zhaoyang Liu, Yukun Chen, Zihao Zhang, Yuji Huang, Yuling Lin, Zhongxiong Lai

**Affiliations:** Institute of Horticultural Biotechnology, Fujian Agriculture and Forestry University, Fuzhou 350002, China; 17805953105@163.com (N.T.); zcynhba@163.com (C.Z.); xuxq0921@163.com (X.X.); 17711828555@163.com (Z.Z.); 18636377888@163.com (J.L.); 13572541682@163.com (Z.L.); cyk68@163.com (Y.C.); zhangzihao863@126.com (Z.Z.); yjhuang2004@163.com (Y.H.); buliang84@163.com (Y.L.)

**Keywords:** *Musa acuminata*, *DWARF53*, dwarf, strigolactone, gibberellin, expression pattern

## Abstract

Dwarfing is one of the common phenotypic variations in asexually reproduced progeny of banana, and dwarfed banana is not only windproof and anti-fallout but also effective in increasing acreage yield. As a key gene in the strigolactone signalling pathway, *DWARF53* (*D53*) plays an important role in the regulation of the height of plants. In order to gain insight into the function of the banana *D53* gene, this study conducted genome-wide identification of banana *D53* gene based on the *M. acuminata*, *M. balbisiana* and *M. itinerans* genome database. Analysis of *MaD53* gene expression under high temperature, low temperature and osmotic stress based on transcriptome data and RT-qPCR was used to analyse *MaD53* gene expression in different tissues as well as in different concentrations of GA and SL treatments. In this study, we identified three *MaD53*, three *MbD53* and two *MiD53* genes in banana. Phylogenetic tree analysis showed that D53 Musa are equally related to D53 Asparagales and Poales. Both high and low-temperature stresses substantially reduced the expression of the *MaD53* gene, but osmotic stress treatments had less effect on the expression of the *MaD53* gene. GR24 treatment did not significantly promote the height of the banana, but the expression of the *MaD53* gene was significantly reduced in roots and leaves. GA treatment at 100 mg/L significantly promoted the expression of the *MaD53* gene in roots, but the expression of this gene was significantly reduced in leaves. In this study, we concluded that *MaD53* responds to GA and SL treatments, but “Yinniaijiao” dwarf banana may not be sensitive to GA and SL.

## 1. Introduction

The *D53 (DWARF53)* gene was originally discovered in short stem dominant mutant rice, in which the *D53* gene was dominantly mutated, and the mutated protein could not be degraded by strigolactone, resulting in a dwarf multiple tiller phenotype [1]. With increasing research, the *D53* gene is considered to be a key gene in strigolactones (SLs) signalling. In rice, the D53 protein interacts with transcription factors, as well as the transcriptional co-repressor proteins TOPLESS (TPL) and TPL RELATED (TPR), to repress the expression of downstream genes when SLs are deficient or at very low concentrations [1]. Whereas, when the concentration of SLs is sufficient to activate the SL signalling pathway, the D14 protein recognises and binds SLs and recruits the SCF (ASK1-CULLIN-F-BOX) complex and D3 protein to further form an SCF–D14 complex. The D3 protein specifically recognises and binds to the SLs signalling inhibitor D53, forming the D53–D14–SCFD3 protein complex. D53 is modified by ubiquitin-conjugating enzyme E2 and degraded by the 26S proteasome to activate the expression of downstream target genes [2]. In dicotyledonous plants, such as *Arabidopsis thaliana*, it was found that the *D53* gene (the immediate homologue of *SMXL6/7/8* in *Arabidopsis*) not only acts as a repressor to inhibit the expression of downstream genes, but *SMXL6* can also directly bind to the promoter of *SMXL6/7/8* and regulate its transcription, thus acting as a transcription factor in SL signalling [3]. *D53/SMXLs* genes have been identified and functionally analysed in a variety of species, such as rice [1], maise [4], *Arabidopsis* [5], cotton [6] and poplar [7], but no systematic identification has been conducted in banana.

Both endogenous and exogenous signals regulate plant growth and development, and hormones, which are one of the major endogenous signals in plants, can respond rapidly to external stimuli [8]. Strigolactones (SLs) are carotenoid-derived terpenoid lactones that are produced in plant roots and transported up the stem to various parts of the plant, regulating multiple stages of plant growth and development [9]. In recent decades, tremendous progress has been made in research on the biosynthesis, signalling and biological functions of SLs. SLs were first found in the seeds of striga hermonthica parasiticum in the roots of cotton and have been shown to play important roles in promoting seed germination, mycelial branching of arbuscular mycorrhizal fungi [10,11], regulation of plant branching [12], flowering period and secondary growth and enhancement of drought and salt tolerance in plants [13]. Among them, five major enzymes are involved in the biosynthesis of SLs, namely D27 (DWARF27), CDD7, CCD8, MAX1 and LOB, and three enzymes are involved in signal transduction, namely Clp proteins D53 (DWARF53)/SMXL6/7/8, D14 (DWARF14) and D3 (DWARF3)/MAX2 [14]. The rice *D10* gene encodes the carotenoid cleavage dioxygenase CCD8, a direct homologue of *Arabidopsis MAX4* [15], and when this gene is mutated, plants exhibit reduced plant height and increased tillering [16]. In addition, *D17* encodes the carotenoid cleavage dioxygenase CCD7, whose partial loss of function leads to dwarfing, multiple tillering and increased yield in rice [17]. D14 is the SL receptor, and *d14* mutants have increased tillering and reduced plant height [18]. D3 is involved in the degradation of inhibitory proteins in the SL signalling pathway, and *Arabidopsis d3* (*max2*) mutants exhibit plant dwarfing and reduced adventitious roots [19].

Gibberellic acid (GA) is also a widely available phytohormone that plays an important role in promoting seed germination, flowering and fruit growth, as well as regulating plant flower development and plant height [20]. Plants defective in GA synthesis or signalling are characterised by plant dwarfism, dark green leaves, stunted growth and reduced seed production [21]. Several key genes for GA signalling have been identified. The binding of GAs to the GA receptor GID1 drives the interaction between GID1 and DELLA, which ultimately leads to the degradation of the DELLA protein by the 26S proteasome [22]. A total of three *GID1* genes were identified in *Arabidopsis*, and it was found that *GID1* single mutants did not have a significant GA-insensitive phenotype, but triple mutants exhibited plant dwarfism and impaired growth and development [23]. *DELLA* belongs to the *GARS* family of transcription factors [24], and when the structure of the protein is altered so that it cannot sense GA signals, resulting in an inability to be degraded, it leads to shortened internodes and plant dwarfism. F-box proteins can interact with target proteins to ubiquitinate them and then degrade them via the 26S proteasome pathway. It was found that DELLA proteins increased, and plants were dwarfed when the *Arabidopsis AtSLY1* gene was functionally deficient [25], whereas DELLA proteins were significantly reduced when the *AtSLY1* gene was overexpressed, and some of the plant heights of the plants were rescued [26]. The *GID2* mutation in rice also results in plant dwarfism accompanied by insensitivity to GAs and reduced fertility [27]. GA-induced degradation of DELLA protein is considered to be a key step in the GA signalling pathway.

The molecular mechanisms of GA and SL signalling are very similar, both acting after hormone-induced protein hydrolysis, and both hormone receptors D14 and GID1 belong to the α/β hydrolase family [28]. It is, therefore, speculated that there may be an interaction between GA and SL. It has been reported that in rice, GAs negatively regulate SL synthesis in plants, that SL levels are higher in GA-insensitive mutants [29] and that D14 can be combined with SLR1 in the presence of SLs [30]. However, studies in pea indicated that the effect of SLs in promoting internode elongation was not affected by GAs [31]. Whether *D53*, an important gene in the SL signalling pathway, responds to GA in banana is unclear.

Banana (*Musa* spp.) is a monocotyledonous plant of the genus *Musa* in the family Musaceae, which is popularly known for its nutrition and taste and is an important economic and food crop worldwide [32]. In the production process, due to the banana plant’s tall, heavy crown and poor wind resistance, it is prone to serious breakage and collapse when it encounters typhoons or tropical storms, causing the banana industry to suffer significant losses [33]. Dwarf varieties of banana plants are short and strong, which not only improves wind resistance but also facilitates cultivation and management [34]. At the same time, it can shorten the growth cycle and effectively improve the mu yield of bananas [35]. Therefore, it is important to study the dwarfing mechanism of banana for the long-term development of the banana industry. *D53/SMXLs*, as a key repressor of the SL signal transduction pathway, plays an important role in SL-mediated plant dwarfing. In order to further investigate the mechanism of the *D53* gene in dwarf banana, this study was based on genomic data from three species of *M. acuminata*, *M. balbisiana* and *M. itinerans*, and the genome-wide identification of the banana *D53* gene. We also investigated the effects of plant height-related hormones GR24 (synthetic SL analogue) and GA on the plant height of “Yinniaijiao” dwarf banana (*Musa* spp. AAA group, a dwarf cultivar from variation via tissue culture) and the expression of its *MaD53* gene, and to provide a reference for further investigation of the function of banana *D53* gene and the mechanism of banana dwarfing.

## 2. Results and Analysis

### 2.1. Identification of D53 Gene and Analysis of the Protein in Banana

Through protein Blast and structural domain prediction analysis, 3 *MaD53*, 3 *MbD53* and 2 *MiD53* genes were finally identified in the genomes of *Musa acuminata*, *M. balbisiana* and *M. itinerans*, respectively. Based on the position information of each member in the genome, the three *MaD53* were named *MaD53-1–MaD53-3*, and the naming principles of *MbD53* and *MiD53* were consistent with those of *MaD53*.

The analysis of the proteins revealed that MaD53, MbD53 and MiD53 proteins had small differences in the number of amino acids (Table 1). Their mean average values are 1175, 1177 and 1057, respectively, with molecular weights ranging from 115,061.28 to 130,219.95 Da and isoelectric points ranging from 5.79 to 6.12. In addition, the instability coefficients of banana D53 proteins were all greater than 40, and the average hydrophilicity coefficients were all negative, indicating that all were unstable hydrophilic proteins. Subcellular localisation prediction showed that all of them were localised to the nucleus, except MaD53-3 and MbD53-3, which were localised to the chloroplasts.

### 2.2. Phylogenetic Analysis of D53

Calculation of the similarity between the D53 sequences of the three genomes of banana, as well as the D53 similarity between banana and other monocotyledonous plants, showed that the similarity between MaD53, MbD53 and MiD53 sequences ranged from 49.7648 to 97.1477 (Appendix A). The similarity between banana D53 and other monocotyledon D53 sequences ranged from 22.2123 to 42.8164, with MiD53-1 and DcD53 having the highest similarity of 42.8164.

To investigate the phylogenetic relationship of the D53 gene, a phylogenetic tree was constructed by MEGA6.06 for D53 proteins of 40 species, including banana, rice, *Arabidopsis*, maise and so on (Figure 1). The D53 proteins of the three genomes in banana clustered into a single unit with an equal evolutionary relationship to the D53 proteins of the Asparagales and Poales, such as *Phalaenopsis equestris* PeD53, *Dendrobium catenatum* DcD53, wheat (TaD53, TdD53, TuD53), sorghum SbD53, maise ZmD53 and rice OsD53. Currently, the *D53* gene has been found only in angiosperms, not in algae, mosses, ferns or gymnosperms, and only the paralogous homologues of *D53*, such as *SMAX1* and *SMXL2*, have been found. It suggested that the *D53* gene originated in angiosperms and evolved by doubling genes such as *SMXL1*, *SMXL2* and so on.

Two or more *D53* genes are commonly found in dicotyledons, which cluster into two evolutionary branches each. In contrast, *D53* genes in monocotyledons are clustered into only one branch, and only one *D53* gene has been found in the basal angiosperms *Zostera marina* (annotated D53), *Amborella trichopoda* and *Nymphaea colorata* (annotated *SMXL7*). It suggested that at an early stage, the *D53* gene underwent a gene doubling event only in the dicot ancestor but not in the monocot ancestor. However, two or more *D53* genes were found in some monocotyledonous plants, such as rice, wheat, sorghum and *Musa acuminata Colla*, suggesting that some monocotyledonous plants underwent a recent *D53* doubling event within the family or genus.

### 2.3. Chromosomal Localisation and Collinearity Analysis of Banana D53 Gene

The chromosomal localisation of the banana *D53* gene is shown in Figure 2A. Both *MaD53* and *MbD53* genes are localised on two chromosomes, chr 7 and 10, where *MaD53-1* and *MbD53-1* are located on chr 7, and *MaD53-2*, *MaD53-3*, *MbD53-2* and *MbD53-3* are all located on chr 10. *M. itinerans* assembled only to the scaffold (S) level, with *MiD53-1* and *MiD53-2* localised on S621 and S671, respectively.

To investigate the gene duplication events of banana *D53*, the *D53* genes of banana, rice and *Arabidopsis* were analysed for collinearity. The results showed that no collinearity existed in both banana *MaD53* and *MiD53*, but collinearity existed between *MbD53* genes with two homologous gene pairs. The collinearity analysis of *D53* genes in all three genomes of banana revealed that *MaD53* found homologous genes in *MbD53,* and *MiD53* found homologous genes in *MaD53*. There were seven homologous gene pairs between *MaD53* and *MbD53*, three homologous gene pairs between *MaD53* and *MiD53* and two homologous gene pairs between *MbD53* and *MiD53*, suggesting that the *D53* gene was conserved in banana. In addition, *MaD53* also has homologous genes in *Arabidopsis thaliana* and rice, and *MaD53-1* has collinearity with two *OsD53* and one *AtD53* gene, respectively (Figure 2B).

The nonsynonymous substitution (Ka) and synonymous substitution (Ks) values of the homologous gene pairs of banana *D53* were calculated to analyse whether the gene was subjected to natural selection pressure during the evolutionary process and to trace its duplication time (Table 2). The Ka/Ks values of the segmentally duplicated gene pairs in the banana *D53* gene were all less than 1, indicating that the banana *D53* gene was subjected to purifying selection during the evolutionary process. The prediction of the replication time of the banana *D53* gene showed that the *MiD53-2* and *MaD53-1* genes had the earliest replication time of about 57.2744 million years ago, followed by *MiD53-2* and *MbD53-1*, with a replication time of about 57.0194 million years ago. It was hypothesised that the *MaD53-1* and *MbD53-1* genes might have both been generated by *MiD53-2* gene duplication.

### 2.4. Gene Structure and Protein Structural Domain Analysis of D53 in Banana

The exon–intron structure of the banana *D53* gene was analysed, and it was found that the *MiD53* gene had the highest number of exons and introns (Figure 3), about twice as many as *MaD53* and *MbD53*. *MaD53* and *MbD53* had only three exons and two introns, except for *MbD53-2,* which had four exons and three introns.

The results of protein structural domain prediction showed that MaD53, MbD53 and MiD53 proteins all contain RecA-like_ClpB_Hsp104-like, P-loop_NTPase superfamily, ClpA, ClpA superfamily and AAA_2 structural domains. It was also found that clpC and clpC superfamily structural domains were present in other D53 proteins besides MaD53-1 and MbD53-1.

Analysis of the 25 conserved motifs of the banana D53 protein revealed that MaD53-1, MaD53-2, MbD53-1 and MbD53-2 all contain 25 motifs. In addition, MaD53-3 and MbD53-3 were missing motif 24, MiD53-1 was missing motif 16 and motif 23 and MiD53-2 was missing motif 23. The high sequence and structural similarity of banana D53 proteins suggested that they may be functionally conserved.

### 2.5. Analysis of Promoter Cis-Acting Elements and Transcription Factor Binding Sites of D53 Gene in Banana

#### 2.5.1. Analysis of Promoter Cis-Acting Elements

Analysis of *cis*-acting elements in the 1500 bp region of the banana *D53* gene promoter, classification and counting of *cis*-acting elements with different functions and their site numbers and the results are shown in Figure 4. Multiple light-responsive elements, hormone-responsive elements and response elements related to stress or plant growth and development are present in the banana *D53* gene promoter. It indicates that the *D53* gene is regulated by various factors and may be involved in the regulation of a variety of physiological processes during banana growth and development.

The banana *D53* gene has the highest number of light-responsive elements in the promoter, ranging from 4 to 15, and the highest number was found in *D53-2* in all three genomes. The banana *D53* gene contains five hormone response elements, abscisic acid (ABA), growth hormone (AUX), gibberellin (GA), methyl jasmonate (MeJA) and salicylic acid (SA). Each *D53* gene has an ABA response element, and all contain at least three hormone response elements, suggesting that the *D53* genes in bananas may be interlinked with other hormones and work together to regulate plant growth and development. Some growth and development-related response elements were also found in the *D53* genes, such as endosperm expression (GCN4_motif), circadian control (circadian), meristem expression (CAT-box) and zeatin metabolism regulation (O2-site). It suggested that the banana *D53* genes may be involved in the regulation of circadian rhythm, protein metabolism and other life processes. A variety of adversity stress response elements were also present in the promoter of banana *D53* gene, including anaerobic-induced (ARE), anaerobic-specific inducibility (GC-motif), low-temperature (LTR), and MYB-binding sites in drought, which suggested that *D53* gene may play a role in abiotic stress processes. In addition, more stress-responsive elements were present in the *MbD53* and *MiD53* genes compared with the *MaD53* gene, indicating that the *MbD53* and *MiD53* genes may play a more important role in banana’s resistance to stresses.

#### 2.5.2. Transcription Factor Binding Site Analysis

In order to investigate the interaction of transcription factors with banana *D53* gene, the transcription factor binding sites (TFBS) within the 1500 bp region of its promoter were categorised and counted. A total of nine transcription factor (TF) families (BBR-BPC, AP2, C2H2, ERF, G2-like, MIKC_MADS, Nin-like, TALE, TCP) were found in the promoter of the banana *D53* gene. Of these, seven are *MaD53-1*, *MbD53-1* and *MiD53-1*; six are *MaD53-3*, *MbD53-2* and *MiD53-2;* and five are *MaD53-2* and *MbD53-3*. BBR-BPC had the highest number of binding sites, with 134, 90 and 43 in the *MaD53*, *MbD53* and *MiD53* genes, respectively, present in each banana *D53* gene. This was followed by MIKC_MADS and TALE, with TALE present in every *D53* gene and MIKC_MADS present in all *D53* genes except *MaD53-2*. The type, number and distribution of transcription factors on different *D53* genes in banana were also different, such as G2-like, which was only present in *MaD53-1*, *MbD53-1* and *MiD53-1* genes; Nin-like, which was only present in *MaD53-2* and *MbD53-2* genes; C2H2, which was only present in *MaD53-2* gene. The largest type and number of transcription factors in banana was the *MaD53* gene, followed by *MbD53*, and MiD53 was the least. Therefore, it is hypothesised that banana *D53* genes play multiple roles in the growth, development and fruiting process of banana, and different *D53* genes may play different roles.

### 2.6. Analysis of Transcriptome Data of MaD53 Gene under Different Treatments

To understand the expression of the *MaD53* gene under various abiotic stresses, the FPKM values of the *MaD53* gene were extracted from the transcriptomic data of high temperature “Tianbaojiao”, different low temperatures of “Sanmingyeshengjiao” and “Grande Naine” Cavendish osmotic stress to plot a heatmap (Figure 5). Under different low-temperature treatments, the expression of *MaD53-1* and *MaD53-2* decreased and then increased, reaching a minimum at 4 °C, and the expression of *MaD53-3* decreased continuously. Compared with the 28 °C control, the expression of *MaD53* was substantially decreased under high-temperature treatments. However, the expression of *MaD53* was only slightly up-regulated under osmotic stress. It can be seen that both high and low-temperature stresses have a large effect on the expression of *MaD53-1* and *MaD53-2*, while *MaD53-3* may have a smaller effect because of its relatively low expression in the plant.

### 2.7. Expression Analysis of MaD53 Gene in Different Tissue Parts of “Yinniaijiao” Dwarf Banana

In order to investigate the specific expression pattern of *MaD53* genes in “Yinni Dwarf”, the expression of three *MaD53* genes was analysed in different tissues using *UBQ2* as an internal reference gene (Figure 6). The three *MaD53* genes were expressed in banana roots, pseudostems and leaves, and they were widely involved in the growth and development of banana. The expression of *MaD53* genes and their expression patterns were different in different parts of banana, among which *MaD53-1* was significantly higher in pseudostem and leaves than in roots. The expression of *MaD53-2* was highest in roots, followed by leaves and, finally, pseudostems. *MaD53-3* was not highly expressed in roots, pseudostems and leaves, and its expression was significantly lower in both pseudostems and leaves than in roots.

### 2.8. Expression Pattern Analysis of MaD53 under Hormone Treatment

Several studies have shown that the *D53* gene in plants responds to SL treatment, but whether *MaD53* expression in banana is regulated by SLs has not been reported. Therefore, in this study, different concentrations of GR24 solution were applied to the “Yinniaijiao” dwarf banana to investigate the effect of exogenous GR24 treatment on the height of banana plants and the expression of *MaD53* gene (the internal reference gene was *CAC*). Plant height of banana was not significantly increased by 5 μM and 10 μM GR24 treatments compared to control (*p* < 0.05) (Figure 7A,B and Table 3). The expression of *MaD53* after GR24 treatment was different in leaves and roots (Figure 7C,D), but most of them were lower than the control. In roots, the expression of *MaD53-1* and *MaD53-3* decreased with the increase in treatment concentration, and the expression of *MaD53-2* was highest at 5 μM and significantly lower than the control at 10 μM. In leaves, the expression of all *MaD53* genes decreased and then increased, with the highest expression in control and the lowest at 5 μM treatment. The above results indicated that all *MaD53* genes responded to GR24 treatment.

In order to investigate whether GA treatment can affect the plant height and *MaD53* gene expression of “Yinniaijiao” dwarf banana, the present study was carried out with different concentrations of GA, and the expression of *MaD53* gene was detected (the internal reference gene was *UBQ2*). Plant height of banana was not significantly increased in different concentrations of GA treatments compared to the control, but the expression of *MaD53* was significantly changed (Figure 8A,B and Table 3). The expression of the three *MaD53* genes in roots showed the same trend of decreasing (Figure 8C,D), then increasing and then decreasing, with the highest expression at 100 mg/L and a significant decrease at 200 mg/L, which was similar to the trend of plant growth rate. In leaves, the expression of *MaD53* genes in the treatment groups was all significantly lower than that in the control group. Among them, the expression of *MaD53-1* and *MaD53-2* firstly increased and then decreased, while the expression of *MaD53-3* continuously decreased. The above results suggested that exogenous GA treatment affected the expression of *MaD53*, a key gene for SL signalling in banana, and that the effect had tissue and concentration variability.

## 3. Discussion

### 3.1. The Banana D53 Gene Evolved with a Gene Doubling Event with a Loss Event

Phylogenetic tree analysis of the D53 protein revealed some important phenomena in the origin and evolution of the *D53* gene in the plant kingdom. One *D53* gene (annotated as *SMXL7*) was present in the basal angiosperms *Amborella trichopoda* and *Nymphaea colorata*, whereas no *D53* gene was found in algae, mosses, ferns and gymnosperms, and there were only its paraphyletic homologous genes *SMAX1*, *SMXL2* and so on. This suggested that the *D53* gene originated in angiosperms, presumably as a result of a gene doubling event in the ancestral genes of *SMAX1* and *SMXL2*. This doubled gene was mutated to form the *D53* gene, the *D53* (*SMXL7*) gene in early angiosperms, which was retained by natural selection in the evolutionary process. The *D53* gene doubling event did not occur in the ancestors of monocotyledonous plants, but some monocotyledonous plants underwent a recent doubling event of the *D53* gene within the family or genus. Thus, some monocotyledonous plants have two or even more *D53* genes, such as *Musa acuminata Colla*, *Oryza sativa Japonica* and wheat. Doubling events of the *D53* gene occurred in early dicot ancestors after the separation from the basal angiosperms, so that more than one *D53* gene is commonly present in dicotyledons. In addition, some other dicotyledons have also undergone recent doubling events of the *D53* gene within the family, such as *Arabidopsis thaliana* in the Cruciferae, peanut in the Leguminosae, apple in the Rosaceae and tobacco in the Solanaceae. Some of their *D53* genes first clustered into a single unit and then collocated with *D53* genes in other species.

It has been pointed out that three whole-genome duplication (WGD) events occurred in banana during the evolutionary process, namely the α, β (about 70 million years ago) and γ events (about 100 million years ago) [36]. The duplication event between banana *MaD53* and *MbD53* genes occurred from 3.3939–56.9012 Mya after the α event, while the A and B genomes diverged at a very close time of about 5.4 Mya [37]. It is hypothesised that the banana A genome underwent a gene-doubling event when it underwent the three WGD events, which resulted in three *MaD53* genes in the A genome. The A and B genomes had begun to diverge by the time the *MaD53* and *MbD53* gene duplications occurred, and therefore, there were also three *MbD53* genes in the B genome after complete divergence. *Musa itinerans* is the same AA-type diploid wild species, and the duplication between *MaD53* and *MiD53* genes was also after three WGD events, but only two *MiD53* genes exist in its genome at present, and it is presumed that the *MiD53* gene was doubled and then lost. The same loss of the *D53* gene occurred in both monocotyledonous and dicotyledonous plants. *Aquilegia coerulea* is a tetraploid plant whose ancestral palaeotetraploidisation occurred after the divergence of the columbine from the opium poppy [38]. There are eight *D53* genes in the opium poppy, whereas only one *D53* gene is found in the columbine, suggesting that a loss event of the *D53* gene in the columbine may have occurred, as well.

### 3.2. MaD53 Gene May Have Evolved Functionally Differently in Banana

The *D53* gene plays an important role in SL signalling as a key repressor of the SL signalling pathway. There are three *D53* immediate homologs in *Arabidopsis*, *AtSMXL6/7/8*, which all function as repressors in the SL pathway but have their own unique functions. In terms of regulating branching, the *smxl6/7/8* single mutant and *smxl6/7*, *smxl6/8*, *smxl7/8* double mutants showed similar phenotypes to the wild type, whereas the number of secondary shoots of the *smxl6/7/8* triple mutant was significantly less than that of the wild type. It suggested that *SMXL6*, *SMXL7* and *SMXL8* had a redundant function in promoting the growth of axillary buds in *Arabidopsis* [39]. In addition, the number of branches in the *smxl7max2* mutant was much less than that in the *smxl6max2* and *smxl8max2* mutants, suggesting that although *SMXL6/7/8* can regulate *Arabidopsis* branching, *SMXL7* plays a major role, which may be related to the fact that *SMXL7* is mainly expressed in axillary branches [5]. Furthermore, it was found that *SMXL6*, in addition to acting as a repressor of the SL pathway, can also act as a transcript that directly binds to the promoter of *AtSMXL6/7/8* and regulates its transcription.

Analysis of *MaD53* gene expression in several tissues of “Yinniaijiao” dwarf banana revealed that *MaD53-1*, *MaD53-2* and *MaD53-3* were expressed in roots, pseudostems and leaves and were widely involved in the growth and development of banana, but the dominant sites of expression of each gene were different. Among them, the expression of *MaD53-1* was higher in leaves and pseudostems and significantly higher than in roots. *MaD53-2* was mainly expressed in roots and leaves, with significantly lower expression in pseudostems than in other tissues. The expression of *MaD53-3* was the highest in roots and significantly higher than in pseudostems and leaves. *MaD53-1* is hypothesised to function mainly in leaves and pseudostems, *MaD53-2* mainly in leaves and roots and *MaD53-3* mainly in roots. The functions of plant genes are, to some extent, regulated by their promoter *cis*-acting elements and transcription factor binding sites [40]. It was found that the number of hormone-responsive elements, as well as the types of stress, growth and development-responsive elements in *MaD53-1,* were much less than those in *MaD53-2* and *MaD53-3*, while the types and numbers of transcription factor binding sites in *MaD53-2* were much less than those in *MaD53-1* and *MaD53-3*. Taken together, it is hypothesised that the function of the *MaD53* gene has evolved in different directions. In addition, the expression trends of *MaD53-2* and *MaD53-3* were found to be more similar, with both having the highest expression in roots, which was quite different from *MaD53-1*, suggesting that the functions of *MaD53-2* and *MaD53-3* may be more similar. Strigolactones are a class of terpene lactones, which are mainly synthesised in roots and transported up the stem to various parts of the aboveground, and then play a role in multiple stages of plant growth and development. *MaD53-1* was mainly expressed in the aboveground part of the plant, which coincided with the function of SLs in the plant, and thus, it was hypothesised that *MaD53-1* played the main function in the SL pathway.

### 3.3. The MaD53 Gene in “Yinniaijiao” Dwarf Banana Responds to GA and SL but Might Not Regulate Plant Height

Strigolactone has an important role in regulating the formation of aboveground plant phenotypes, including the number of branches, leaf shape, leaf colour, the height of the main stem and thickness, and has been investigated in a variety of plant species. Plant height, one of the important agronomic traits of crops, dwarfing the plant under the premise of ensuring the biological yield of the crop can not only effectively improve the ability of resistance to collapse [41] but also make full use of the land space, improve the utilisation of light energy and increase the yield per unit area [42]. Studies in recent years have shown that GA and SL biosynthesis and signal transduction are usually related to plant height [43]. Studies have shown that spraying GA3 significantly promotes kale plant growth [44]. In rice, the “green revolution” gene *sd1* regulates rice plant height by participating in GA biosynthesis [45], and the wheat semi-dwarfing genes *Rh1* and *Rh2* are involved in GA signalling [46]. The SL synthesis pathway mutant *d10* and *d17* and the signalling pathway mutants *d13*, *d4* and *d53* all showed dwarfing and multiple tillering traits, and the application of GR24 alleviated the dwarfing and multiple tillering phenotypes of *d10* and *d17* but not those of *d3*, *d14* and *d53* [16,47,48], and significantly down-regulated the expression of their *D53* gene [2]. In apple, lower concentrations of GA did not promote plant height and internode elongation in A1d (dwarf plants formed by a GA signalling mutation), but it responded to high concentrations of GA [49]. In this experiment, GA and GR24 were sprayed on “Yinniaijiao” dwarf banana, and it was found that GA and GR24 could not significantly increase plant height (*p* < 0.05), but the expression of *MaD53* gene was mainly down-regulated in roots and leaves after the treatments. This suggested that although the *MaD53* gene responded to GA and SL treatments, “Yinniaijiao” dwarf banana may not be sensitive to GA or GR24. In this regard, the following speculations were made: (i) The concentration of the hormone used was not high enough to reach the threshold for significant plant height promotion. (ii) The GA or SL signalling genes were mutated, resulting in their inhibitory proteins not being degraded and the dwarfing phenotype not being alleviated. In addition, 100 mg/L GA treatment promoted the expression of the *MaD53* gene in roots, but the expression of this gene was significantly reduced in leaves. The *D53* gene is a repressor of the SL signalling pathway, and elevated expression of this gene inhibits plant growth. It is, therefore, hypothesised that the rise in *MaD53* gene expression in roots at this concentration inhibited root growth and thus resulted in failure to promote plant height.

Since the molecular mechanisms of GA and SL signalling are very similar [50], and the receptor proteins both belong to the α/β-hydrolase family [51], several studies have indicated the existence of interactions between GA and SL, but their interactions seem to be different in different species. In jatropha, SL treatment up-regulated the expression of *NAC* genes, contrary to the results of GA treatment [52]. In rice, GA treatment decreased the expression of SL synthesis genes, and thus SL levels in vivo, and the regulation was induced through the GID-DELLA signalling pathway [53]. GR24 treatment not only binds D14 and D53, the key genes of the SL signalling pathway but also binds D14 and the DELLA protein SLR [30]. In cucumber, the expression of the *CsD14* gene in roots and leaves was suppressed under low-concentration GA treatment and significantly increased under high-concentration GA treatment. In contrast, the expression of the *CsD14* gene was significantly increased by both high and low-concentration treatments in stem and leaf axils [54]. This suggested that the regulation of SL signalling pathways by GAs was tissue- and concentration-specific and varies in different tissues. In this study, we found that the expression of *MaD53*, a key gene of the SL pathway, was significantly changed in both roots and leaves under different concentrations of GA treatment, suggesting an interaction between GA and SL in banana.

## 4. Materials and Methods

### 4.1. Material Handling

One-month-old “Yinniaijiao” dwarf banana (*Musa* spp. AAA group, a dwarf cultivar from variation via tissue culture) transplants with uniform growth were selected for GR24 (SL analogue) and GA treatments. GR24 concentrations were set at 5 μM and 10 μM, GA concentrations were set at 50 mg/L, 100 mg/L and 200 mg/L by spraying method until the leaves and pseudostems were fully wetted, but no droplets fell, and the control group (CK) was sprayed with equal amount of water. Spray every 5 days for a total of 3 sprays. The plant heights were measured after the treatments, and the mean values were taken for statistical analysis. All the treatments were carried out in three biological replications. The first leaf (unfolded leaf), root and different tissue parts (leaf, pseudostem and root) of the control plants were taken and placed in liquid nitrogen for quick freezing and stored in an ultra-low temperature refrigerator at −80 ℃ for total RNA extraction, qRT-PCR expression analysis.

### 4.2. Methods

#### 4.2.1. Identification of Banana D53 Gene and Phylogenetic Tree Construction

Whole genome files, annotation files and protein files of banana *M. acuminata var*. *DH-Pahang* [55], *M. balbisiana var. DH PKW* [37] and *M. itinerans* were downloaded from the Banana Genome Database (http://banana-genome-hub.southgreen.fr, accessed on 10 December 2021). D53 protein sequences of other species downloaded from the NCBI database (https://www.ncbi.nlm.nih.gov/, accessed on 15 June 2022).

Homologous blast of banana protein sequences with rice OsD53 and *Arabidopsis thaliana* SMXL6/7/8 protein sequences by TBtools-II v2.042 [56], respectively, with the parameter threshold E-value ≤ 1 × 10^−5^ and other parameters as default values. The banana D53 sequence was initially screened by two-way Blast of the obtained candidate sequences using the NCBI website. Combined with NCBI-CDD [57] (https://www.ncbi.nlm.nih.gov/Structure/cdd/cdd.shtml, accessed on 3 September 2023) and SMART [58] (http://smart.embl-heidelberg.de/, accessed on 3 September 2023) predictive analyses of protein structural domains, banana *D53* genes finally identified and named with reference to the nomenclature of rice and *Arabidopsis thaliana*.

Phylogenetic trees were constructed using the maximum likelihood (ML) method of the software MEGA6.06 for D53 proteins of several species, including banana and *Arabidopsis thaliana*. The sequence alignment method was Muscle. The running model was Jones–Taylor–Thornton (JTT) + Gamma Distributed (G), with the Bootstrap value set to 1000, and other parameter values were defaulted. Similarity of D53 protein sequences was calculated by DNAMAN 6.0 (Lynnon Biosoft) software.

#### 4.2.2. Chromosomal Localisation and Collinearity Analysis of Banana D53 Gene

Based on the genome files and annotation files of banana, rice and *Arabidopsis thaliana*, TBtools-II v2.042 was used to map the diagram for Chromosomal (chr) localisation and collinearity of the banana *D53* gene and the values of nonsynonymous substitutions (Ka) and synonymous substitutions (Ks) were calculated for *D53* replicated genes in banana. The gene duplication occurrence time (T) was calculated as T = Ks/2λ, where the evolutionary rate (λ) of Musa was 4.5 × 10^−9^ [59].

#### 4.2.3. Structure and Protein Analysis of Banana D53 Gene

ExPASy’s [60] online tool ProtParam (https://web.expasy.org/protparam/, accessed on 15 September 2023) were used to predict the amino acids, molecular weight, theoretical isoelectric point, instability coefficient and average hydrophilicity coefficient of banana D53 protein, and similarity of D53 protein sequences was calculated by DNAMAN 6.0 (Lynnon Biosoft) software.

Subcellular localisation of banana D53 protein was predicted using WoLF PSORT [61] (https://wolfpsort.hgc.jp/, accessed on 15 September 2023). The conserved motifs and structural domains of banana D53 protein were analysed by MEME [62] (https://meme-suite.org/meme/, accessed on 3 September 2023) and NCBI-CDD, respectively, with the number of conserved motif identifications set to 25. The resulting files were submitted to TBtools for visual mapping.

#### 4.2.4. Prediction of *Cis*-Acting Elements and Transcription Factor Binding Sites in the Promoter of the Banana *D53* Gene

TBtools was used to extract 1500 bp upstream of the transcription start site of the *D53* gene from the banana genome file and submitted to PlantCARE [63] (http://bioinformatics.psb.ugent.be/webtools/plantcare/html/, accessed on 4 September 2023) and PlantTFDB [64] (http://planttfdb.cbi.pku.edu.cn/, accessed on 5 September 2023) online websites (with parameters set to *p*-value ≤ 1 × 10^−6^) for prediction of its promoter *cis*-acting elements and transcription factor binding sites. The results obtained were collated and plotted visually using TBtools.

#### 4.2.5. Analysis of Banana *MaD53* Gene Expression Pattern

FPKM (fragments per kilobases per million mapped) values of *MaD53* gene were extracted from the transcriptome data of high temperature in “Tianbaojiao”, low temperature in “Sanmingyeshengjiao” [65] (A wild banana from Fujian), and osmotic stress in “Grande Naine” Cavendish [66], and submitted to TBtools for heatmap plotting and expression analysis.

Primers were designed using Primer Premier 6 for the *MaD53* sequences with target fragment sizes of 80–200 bp, and the primer sequences and annealing temperatures are shown in Table 4. Total RNA was extracted from roots, pseudostems, and leaves of “Yinniaijiao” dwarf banana treated with water (CK), GA, and GR24, respectively, according to the instructions of the FastPure Universal Plant Total RNA Isolation Kit (Vazyme, Nanjing, China). The concentration, purity and integrity of the extracted RNA were determined by 1.0% non-denaturing agarose gel electrophoresis and UV spectrophotometer. The RNA was reverse transcribed into cDNA for qPCR using the Thermo Scientific RevertAid Master Mix Reverse Transcription Kit (Thermo Fisher Scientific, Waltham, MA, USA). *UBQ2* and *CAC* were used as the internal reference genes [67]. Amplification was performed using a Roche LightCycler 480 fluorescent quantitative PCR instrument, and three replicates of each sample were averaged. The relative expression of target genes was calculated using 2^−∆∆CT^, Microsoft Excel 2016 and GraphPad Prism (version 8.0.2 for Windows, GraphPad Software, San Diego, CA, USA, www.graphpad.com, accessed on 19 November 2023) software programs were used for calculation and graphing, and IBM SPSS Statistics for Windows, version 26.0 (IBMCorp., Armonk, NY, USA) was used for significance analysis.

## 5. Conclusions

In this study, 3 *MaD53*, 3 *MbD53* and 2 *MiD53* genes were identified based on the genomic data of *Musa acuminata, M. balbisiana* and *M. itinerans*. The banana *D53* gene promoter has a large number of light-responsive, hormone-responsive and adversity stress-responsive elements, indicating that this gene may have important roles in plant growth and development and adversity stress. The evolutionary tree and collinearity analysis hypothesised that the banana *D53* genes may have undergone gene doubling events and loss events during the evolutionary process. Tissue-specific expression analysis revealed that the three *MaD53* genes were expressed in various tissues, but the site of dominant expression was different for each gene, and it was hypothesised that each *MaD53* gene had a different major site of function. GR24 and GA treatments reduced the expression of the *MaD53* gene but did not significantly affect plant height in the “Yinniaijiao” dwarf banana, so it was hypothesised that “Yinniaijiao” dwarf banana was not sensitive to GA and SL and that *MaD53* might not be able to regulate its plant height.

## Figures and Tables

**Figure 1 plants-13-00458-f001:**
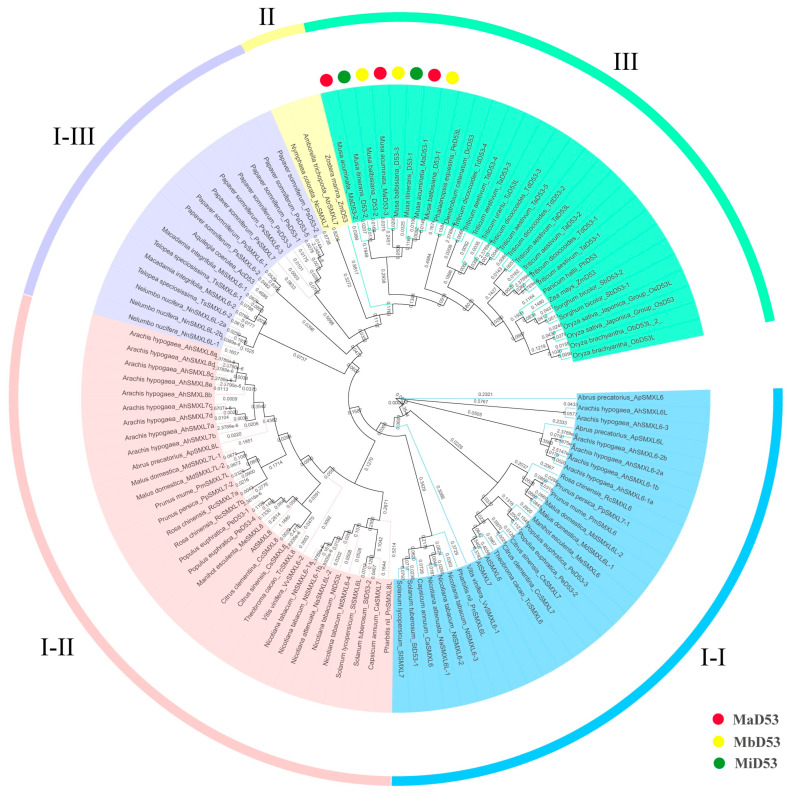
Phylogenetic tree of D53 proteins. Colour blocks I-I, I-II and I-III are all dicot D53 proteins. Colour block II is a basal angiosperm D53 protein, and colour block III is a monocot D53 protein.

**Figure 2 plants-13-00458-f002:**
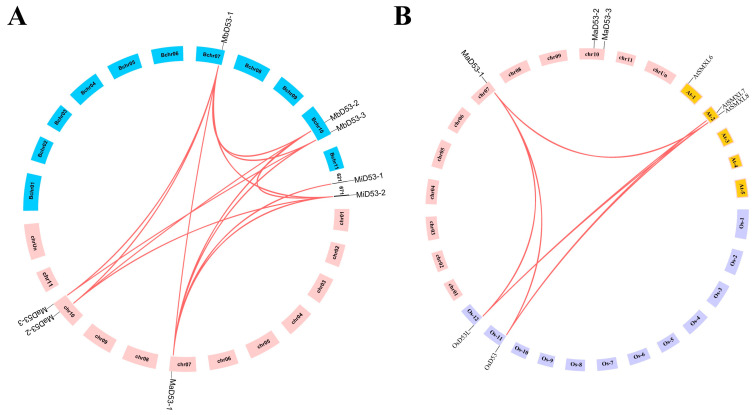
The co-linear distributions of *D53* gene in banana. (**A**) The co-linear distribution of banana *MaD53*, *MbD53* and *MiD53* genes. (**B**) The co-linear distribution of banana, rice and *Arabidopsis D53* genes. The line indicates the segmental replication gene pair of *D53* in banana.

**Figure 3 plants-13-00458-f003:**
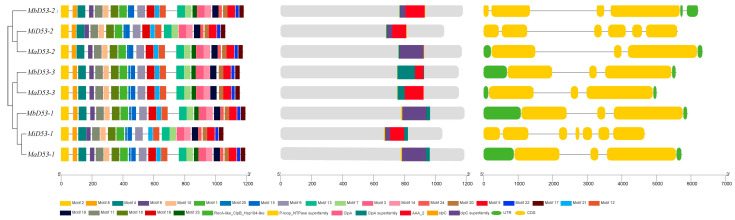
The analysis of gene structures and protein domain of *D53* genes.

**Figure 4 plants-13-00458-f004:**
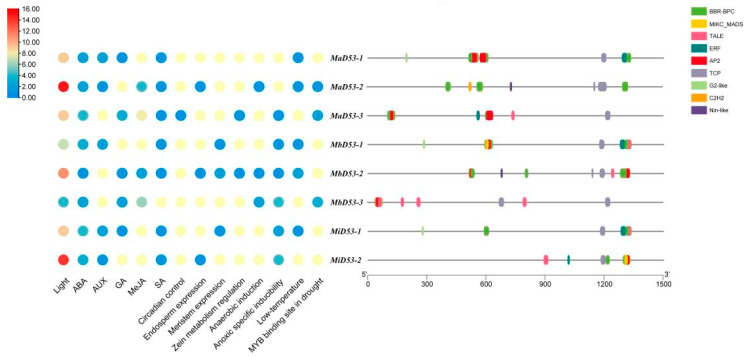
Prediction of *cis*-acting elements (**left**) and transcription factor binding sites (**right**) in the promoters of *D53* in banana.

**Figure 5 plants-13-00458-f005:**
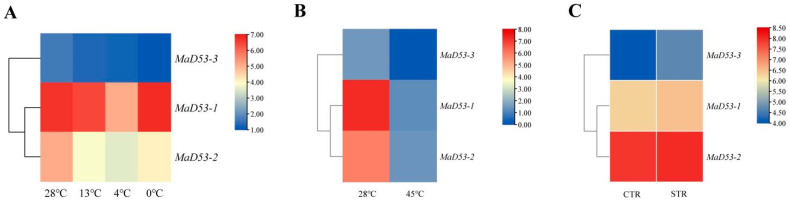
Heatmap of banana *MaD53* gene expression. (**A**) Transcriptome data of ‘Sanmingyyeshengjiao’ banana leaves from different low-temperature treatments and the control at 28 °C. (**B**) Transcriptome data of ‘Tianbaojiao’ banana leaves from high-temperature treatments at 45 °C and the control at 28 °C. (**C**) Transcriptome data of the osmotic stress treatments and the control.

**Figure 6 plants-13-00458-f006:**
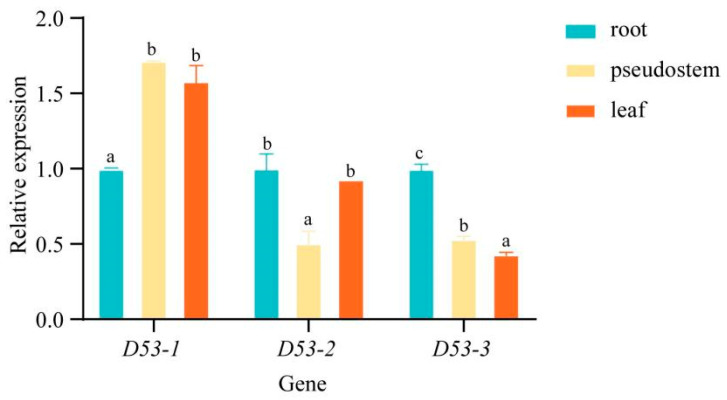
Relative expression levels of *MaD53* genes in different tissues. Lowercase letters indicate significant differences at *p* < 0.05.

**Figure 7 plants-13-00458-f007:**
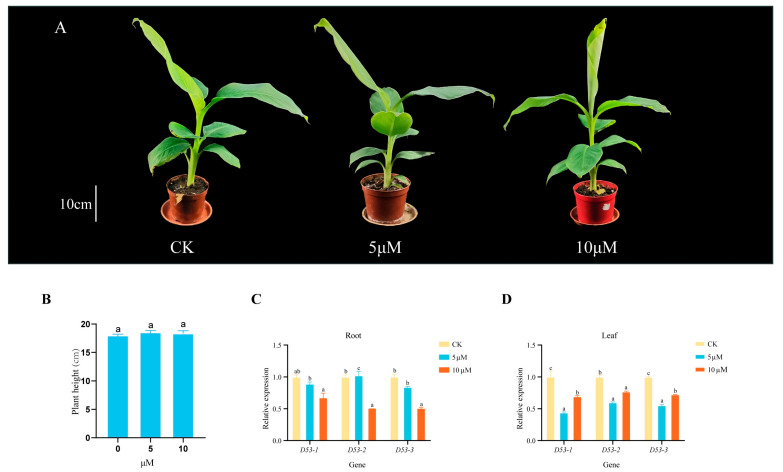
(**A**) “Yinniaijiao” dwarf banana treated with different concentrations of GR24. (**B**–**D**) Growth rate and expression levels of *MaD53* gene in roots and leaves of “Yinniaijiao” dwarf banana under GR24 treatment. Lowercase letters indicate significant differences at *p* < 0.05.

**Figure 8 plants-13-00458-f008:**
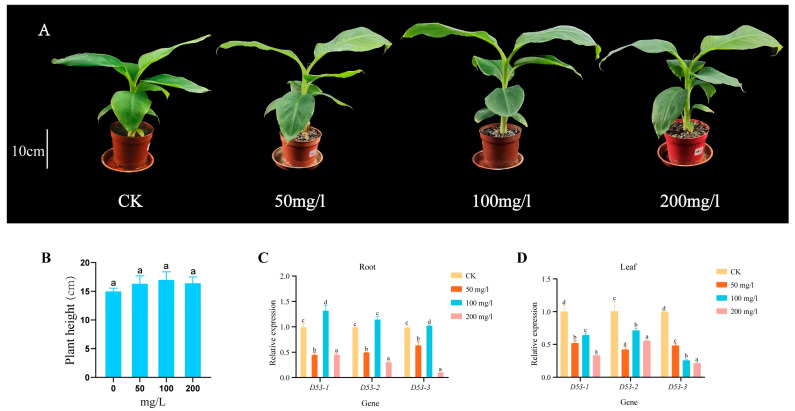
(**A**) “Yinniaijiao” dwarf banana treated with different concentrations of gibberellic acid (GA). (**B**–**D**) Growth rate and expression levels of *MaD53* gene in roots and leaves under different concentrations of GA treatment. Lowercase letters indicate significant differences at *p* < 0.05.

**Table 1 plants-13-00458-t001:** The basic information of *D53* gene in banana.

Gene ID	Gene Name	Size/aa	Molecular Weight/bp	pI	Instability Index	Average Hydrophilic Coefficient	Subcellular Localisation
Ma07_g23940.1	MaD53-1	1192	130,219.95	5.85	53.13	−0.304	nc ^1^
Ma10_g07240.1	MaD53-2	1176	128,235.35	6.1	54.09	−0.351	nc ^1^
Ma10_g29420.1	MaD53-3	1157	126,430.99	5.88	47.92	−0.277	chl ^2^
Mba07_g22340.1	MbD53-1	1193	130,246.76	5.94	53.73	−0.316	nc ^1^
Mba10_g06250.1	MbD53-2	1182	129,064.65	6.12	54.7	−0.327	nc ^1^
Mba10_g25520.1	MbD53-3	1156	126,289.73	5.79	48.58	−0.278	chl ^2^
Mi_g027038	MiD53-1	1051	115,061.28	6.12	53.43	−0.294	nc ^1^
Mi_g027998	MiD53-2	1063	116,132.69	5.99	54.6	−0.366	chl ^2^

^1^ nc: nucleus; ^2^ chl: chloroplast.

**Table 2 plants-13-00458-t002:** *D53* gene replication events in banana.

Gene Name	Gene ID	Gene Name	Gene ID	Ka	Ks	Ka/Ks	Duplication/Mya
MbD53-1	Mba07_g22340.1	MbD53-3	Mba10_g25520.1	0.1269	0.3822	0.332	42.4645
		MbD53-2	Mba10_g06250.1	0.15	0.5026	0.2984	55.8422
		MaD53-1	Ma07_t23940.1	0.0107	0.0305	0.3493	3.3939
		MaD53-3	Ma10_t29420.1	0.1281	0.3644	0.3516	40.4893
		MaD53-2	Ma10_t07240.1	0.1471	0.4955	0.2968	55.0577
		MiD53-2	Mi_g027998	0.1448	0.5132	0.2822	57.0194
MbD53-2	Mba10_g06250.1	MaD53-1	Ma07_t23940.1	0.1521	0.5121	0.2969	56.9012
		MaD53-2	Ma10_t07240.1	0.0137	0.04	0.3414	4.4492
		MiD53-2	Mi_g027998	0.0137	0.0257	0.5315	2.8563
MbD53-3	Mba10_g25520.1	MaD53-3	Ma10_t29420.1	0.0191	0.0456	0.4183	5.0675
		MaD53-1	Ma07_t23940.1	0.1255	0.3827	0.3278	42.5194
MaD53-1	Ma07_t23940.1	MiD53-1	Mi_g027038				
		MiD53-2	Mi_g027998	0.1467	0.5155	0.2846	57.2744
MaD53-2	Ma10_t07240.1	MiD53-2	Mi_g027998	0.0124	0.0285	0.4363	3.1614

**Table 3 plants-13-00458-t003:** Effect of different concentrations of exogenous hormone spraying on plant height of “Yinniaijiao” dwarf banana.

Concentration of Hormone	Plant Height/cm
CK1 (water)	(15.11 ± 0.43) a
GA (50 mg/L)	(16.46 ± 1.23) a
GA (100 mg/L)	(17.13 ± 1.27) a
GA (200 mg/L)	(16.56 ± 0.95) a
CK2 (water)	(17.99 ± 0.41) a
GR24 (5 μM)	(18.53 ± 0.53) a
GR24 (10 μM)	(18.34 ± 0.49) a

Note: Lowercase letters indicate significant differences at *p* < 0.05.

**Table 4 plants-13-00458-t004:** Fluorescent quantitative PCR primer information of *MaD53* gene.

Gene	Primer Sequences (5′→3′)	Size (bp)
MaD53-1	qF: ACACGGAGAGGACCTGCAATC qR: GATACTCTTGCTGGCTGCGGTG	125
MaD53-2	qF: GATTACCACAGCGAGCCAGAG qR: ACAGAGGAAGGTGAAGCGTG	162
MaD53-3	qF: GAATTGCCGGAGGATAGGGG qR: ATTCCATGTAGCTCTGGCGG	153
MaUBQ2	qF: GGCACCACAAACAACACAGG qR: AGACGAGCAAGGCTTCCATT	379
MaCAC	qF:CTCCTATGTTGCTCGCTTATG qR: GGCTACTACTTCGGTTCTTTC	146

## Data Availability

Data are contained within the article.

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
