# Peer review of "Genome-Wide Identification and Expression Analysis of *DWARF53* Gene in Response to GA and SL Related to Plant Height in Banana"

_plants, 2024, doi:10.3390/plants13030458_

Round 1
Reviewer 1 Report
Comments and Suggestions for Authors
Tong et al. conducted this study to identify banana D53 gene based on 13 the M. acuminata, M. balbisiana and M. itinerans genome databases. The banana D53 sequence was initially screened using the NCBI website, then NCBI-CDD were used to identify banana D53 genes. Phylogenetic trees of D53 proteins of several species were obtained. Chromosomal localization and collinearity of D53 gene along with the values of nonsynonymous substitutions and synonymous substitutions were calculated. Using TBtools cis-acting elements and transcription binding sites in the promoter of the banana D53 Gene were identified. MaD53 gene expression patterns at two different temperature and osmotic were identified using the transcriptome data. Primers were designed from MaF53 sequences and used on RNA extracted from RNA was extracted from roots, pseudostems, and leaves of “Yinniaijiao” dwarf banana treated with water (CK), GA, and GR24.
Finally, 3 MaD53, 3 MbD53 and 2 MiD53 genes were identified. The banana D53 gene promoter was characterized. Authors hypothesized that the banana D53 genes may have undergone gene doubling events and loss events during the evolutionary process. Results of this study based on tissue-specific expression analysis revealed that the three MaD53 genes were expressed in various tissues, but the site of dominant expression was different for each gene, and it was hypothesized that each MaD53 gene had a different major site of function. GR24 and GA treatments reduced the expression of MaD53 gene but did not significantly affect plant height in “Yinniaijiao”. Thus it was hypothesised that “Yinniaijiao” dwarf banana was not sensitive to GA and SL, and that MaD53 might not be able to regulate its plant height.
One important aspect I noticed is that I think further information is required to clarify applications of GR24 and GA. Duration of spraying should be explained. Was one spraying used or more than one?
Comments on the Quality of English LanguageThis manuscript needs a considerable amount of work on writing, spacing errors and misspellings.
Author Response
Dear Reviewer,
Thank you for your careful review of the manuscript and put forward many valuable opinions, which is of great help to us to further revise the article. Compared with the review report, we made the following modifications to the manuscript.
Point 1:One important aspect I noticed is that I think further information is required to clarify applications of GR24 and GA. Duration of spraying should be explained. Was one spraying used or more than one?
Response 1: In the paper, we describe the spraying method. (Line 479-481)
We also made some changes to the paper. Please see the revised manuscript for details.
Looking forward to your reply.
Best regards,
Ning Tong
Institute of Horticultural Biotechnology, Fujian Agriculture and Forestry University, Fuzhou, 350002, China
E-mail: 17805953105@163.com

Reviewer 2 Report
Comments and Suggestions for Authors
This manuscript has several strengths that warrant publication. Foremost, is the discovery of three dwarfing genes in three species of banana, and their relative expression in root, pseudostem and leaf tissues, as well as their response to high and low temperature stresses, and lack of response to growth regulation application. Another informative strength is the is the complete historical documentation of dwarfing genes and their mode of action in several important economic genera. Genomic interpretation is informative and basic to understanding key roles of these genes.
The major weakness is lack of attention to specific details in reporting growth responses of various treatments. Specifically, growth rate is not presented accurately. Growth rate implies a quantitative measure of weight, height, leaf area, etc., per a unit of time, such as days weeks, hours, etc. Figures 7 and 8 do not meet these criteria. Relative growth is not well defined.There are also no data to indicate replications, or repeats of the treatments employed. The photos themselves are not convincing without parameters.
The manuscript is very well written.
The following are a few editorial corrections and suggestions:
L16: RtqPCR was used to analyse….
L51: and,
L82: its’ nutritional value and taste …delete tasty taste.
L93: based on genomic data from three species…
L94: delete {were performed)
L118: a instead of an..
L213: It has been suggested (reference)
LL354: It has been suggested (reference).
Author Response
Response to Reviewer 2 Comments
Dear Reviewer,
Thank you for your careful review of the manuscript and put forward many valuable opinions, which is of great help to us to further revise the article. Compared with the review report, we made the following modifications to the manuscript.
Point 1: L16: RtqPCR was used to analyse….
Response 1: In the paper, the sentence has been changed to “RT-qPCR was used to analyse MaD53 gene expression in different tissues as well as in different concentrations of GA and SL treatments”.(Line 16)
Point 2:L51: and,
Response 2: In the paper, the sentence has been changed to “Both endogenous and exogenous signals regulate plant growth and development, and hormones, which are one of the major endogenous signals in plants, can respond rapidly to external stimuli”.(Line 50-52)
Point 3:L82: its’ nutritional value and taste …delete tasty taste.
Response 3: In the paper, We deleted the word “tasty”. (Line 103)
Point 4:L93: based on genomic data from three species…
Response 4: In the paper, “based on three genomic data” has been changed to “based on genomic data from three species”. (Line 114)
Point 5:L94: delete {were performed)
Response 5: In the paper, We deleted “were performed”. (Line 115)
Point 6:L118: a instead of an..
Response 6: In the paper, the word “an” has been changed to “a”. (Line 140)
Point 7:L213: It has been suggested (reference)
Response 7: Sorry, here is speculation based on results obtained and no references. (Line 236)
Point 8:LL354: It has been suggested (reference).
Response 8: Sorry, here is speculation based on results obtained and no references. (Line 377)
Point 9:The major weakness is lack of attention to specific details in reporting growth responses of various treatments. Specifically, growth rate is not presented accurately. Growth rate implies a quantitative measure of weight, height, leaf area, etc., per a unit of time, such as days weeks, hours, etc. Figures 7 and 8 do not meet these criteria. Relative growth is not well defined.There are also no data to indicate replications, or repeats of the treatments employed. The photos themselves are not convincing without parameters.
Response 9: In the paper, based on the content requirements, we modified the calculation to the effect of different hormones on the height of banana plants and added a table to supplement it. (Line 302-324)
We also made some changes to the paper. Please see the revised manuscript for details.
Looking forward to your reply.
Best regards,
Ning Tong
Institute of Horticultural Biotechnology, Fujian Agriculture and Forestry University, Fuzhou, 350002, China
E-mail: 17805953105@163.com
